# Role of Microtubule-Associated Protein 1b in Urothelial Carcinoma: Overexpression Predicts Poor Prognosis

**DOI:** 10.3390/cancers12030630

**Published:** 2020-03-09

**Authors:** Tsu-Ming Chien, Ti-Chun Chan, Steven Kuan-Hua Huang, Bi-Wen Yeh, Wei-Ming Li, Chun-Nung Huang, Ching-Chia Li, Wen-Jeng Wu, Chien-Feng Li

**Affiliations:** 1Graduate Institute of Clinical Medicine, College of Medicine, Kaohsiung Medical University, Kaohsiung 807, Taiwan; u108801005@kmu.edu.tw (T.-M.C.); u8401067@yahoo.com.tw (W.-M.L.); ccli1010@hotmail.com (C.-C.L.); wejewu@kmu.edu.tw (W.-J.W.); 2Department of Urology, Kaohsiung Medical University Hospital, Kaohsiung 807, Taiwan; cnhuang.uro@gmail.com; 3Department of Urology, School of Medicine, College of Medicine, Kaohsiung Medical University, Kaohsiung 807, Taiwan; bewen90@yahoo.com.tw; 4Institute of Biomedical Science, National Sun Yat-sen University, Kaohsiung 80424, Taiwan; ibosaa@mail.nsysu.edu.tw; 5Department of Pathology, Chi Mei Medical Center, Tainan 710, Taiwan; 6Department of Urology, Chi Mei Medical Center, Tainan 710, Taiwan; skhsteven@gmail.com; 7Department of Urology, Ministry of Health and Welfare Pingtung Hospital, Pingtung 900, Taiwan; 8Department of Urology, Kaohsiung Municipal Ta-Tung Hospital, Kaohsiung 801, Taiwan; 9Center for Infectious Disease and Cancer Research, Kaohsiung Medical University, Kaohsiung 807, Taiwan; 10Center for Stem Cell Research, Kaohsiung Medical University, Kaohsiung 807, Taiwan; 11Institute of Medical Science and Technology, National Sun Yat-sen University, Kaohsiung 80424, Taiwan; 12Department of Biotechnology, Southern Taiwan University of Science and Technology, Tainan 71005, Taiwan; 13National Cancer Research Institute, National Health Research Institutes, Tainan 70456, Taiwan

**Keywords:** urothelial carcinoma, transcriptome, microtubule, MAP1B, prognosis

## Abstract

We sought to examine the relationship between microtubule-associated proteins (MAPs) and the prognosis of urothelial carcinoma by assessing the microtubule bundle formation genes using a reappraisal transcriptome dataset of urothelial carcinoma (GSE31684). The result revealed that microtubule-associated protein 1b (*MAP1B*) is the most significant upregulated gene related to cancer progression. Real-time reverse-transcription polymerase chain reaction was used to measure *MAP1B* transcription levels in urothelial carcinoma of the upper tract (UTUC) and the bladder (UBUC). Immunohistochemistry was conducted to detect *MAP1B* protein expression in 340 UTUC and 295 UBUC cases. Correlations of *MAP1B* expression with clinicopathological status, disease-specific survival, and metastasis-free survival were completed. To assess the oncogenic functions of *MAP1B*, the RTCC1 and J82 cell lines were stably silenced against their endogenous *MAP1B* expression. Study findings indicated that *MAP1B* overexpression was associated with adverse clinical features and could independently predict unfavorable prognostic effects, indicating its theranostic value in urothelial carcinoma.

## 1. Introduction

Urothelial carcinoma (UC) is the most common malignancy of the urinary tract and includes UC of the urinary bladder (UBUC) and upper urinary tract (UTUC). UBUC is a major UC, with an estimated 429,800 new cases and 165,100 deaths annually worldwide [1]. When first diagnosed, UBUC presents in most patients as a non–muscle-involved invasive disease with an estimated five-year survival rate of 88%, but this rate dramatically decreases to 15% in patients with tumor metastasis [2]. The prevalence of UTUC accounts for approximately 5% to 10% of all UC cases [3]; however, in Taiwan, the rate of UTUC is as high as 30% of affected cases. Furthermore, there is a slight predominance toward females, and ureteral tumors are attributed to greater than half of all cases of UTUC [4,5].

Transurethral resection of the bladder and radical nephroureterectomy with bladder cuff excision remain the gold-standard treatments in UBUC and UTUC for adequate local tumor control and improved long-term survival. However, despite proper surgical treatment, the mortality rate remains high [2,6,7]. Clinical prognostic factors, such as pathological tumor stage and grade, have diverse impacts in patients with identical findings; therefore, they are insufficient means for detailed risk stratification and are difficult to define before treatment [5].

UBUC staging starts from papillary (Ta) and superficial (T1) stages and extends to muscle-invasive advanced stages (T2–T4). Although the recurrence rate of superficial tumors following surgical resection of the bladder is high, it is associated with a markedly better prognosis than that of muscle-invasive tumors [8]. There is a growing pool of evidence to suggest a pathophysiological distinction exists between superficial and muscle-invasive cases of UBUC [9]. It is also important to distinguish a particular variant that may be associated with the administration of a therapy distinctive from that used in conventional invasive UC [10]. A previous study demonstrated that the gene expression profiles of UC from renal pelvis, ureter and bladder were highly similar, indicating that a common functional molecular pathway likely underlies the carcinogenesis [11]. A larger, follow-up study to elucidate better genomics-based predictors for UC is warranted, the results of which could lead to improvements in neoadjuvant/adjuvant therapy and provide suitable follow-up strategies.

Microtubules are a critical component of the cytoskeleton and are important and indispensable in several cellular processes. They are located throughout the cytoplasm and are dynamically unstable (i.e., coexisting in a state of assembly and disassembly). Microtubule-associated proteins (MAPs) are a large family of proteins involved in microtubule assembly, which is an essential step in stabilizing microtubules. MAPs are divided into two classical families: type I, which includes the MAP1 (MAP1A, MAP1B, and MAP1S) proteins [12] and type II, which includes MAP2, MAP4, and MAPT/TAU proteins [13]. Disrupting microtubule dynamics is one of the most successful and widely considered targets of cancer chemotherapy agents [14,15]. Microtubule agents target the aberrant expression of MAPs in a variety of malignancies, and their resistant phenotypes have been documented. Herein, we aimed to examine the relationship between MAPs and the prognosis of urothelial carcinoma by assessing the microtubule bundle formation genes using a reappraisal transcriptome dataset of urothelial carcinoma (GSE31684). Moreover, to our knowledge, this study is the first to examine *MAP1B* expression and the prognosis and intrinsic biologic aggressiveness of UC.

## 2. Results

### 2.1. MAP1B Is the Most Significantly Upregulated Gene Associated with Microtubule Bundle Formation in UBUC Transcriptomes

The UBUC transcriptome dataset includes 93 tissue samples, with 78 categorized as deeply invasive tissues (pT2–pT4) and 15 categorized as noninvasive or superficial (pTa and pT1) tissues. Metastasis was detected in 28 patients and absent in 49 patients. Through transcriptome profiling, we identified 11 probes spanning six transcripts associated with microtubule bundle formation (GO:0001578). Among these expressed genes, we found that tumors with increased *MAP1B* expression and decreased *MARK4* had a more advanced pT status and a higher incidence of metastatic events (Figure 1A). Our main goal was to find the most significant upregulated genes associated with advanced disease. Therefore, we choose *MAP1B* for further validation. Table 1 shows the *MAP1B* gene (Probe: 226084_at, 214577_at) upregulation with up to 1.2832-, 0.3773- and 0.9436-, 0.3943- fold log ratios in advanced and metastatic UC, respectively. Furthermore, we found through survival analysis that increased *MAP1B* expression was significantly related to poor prognosis in patients with UBUC (Figure 1B). As shown in Figure 1c,d, the *MAP1B* transcripts level was significantly higher among tumors with high pT status (pT2–pT4) than in noninvasive tumors (pTa–pT1) in both the UTUC and UBUC groups (both *p* < 0.01). Our findings indicate that *MAP1B* is associated with tumor aggressiveness.

### 2.2. MAP1B Immunoexpression and Clinicopathological and Genomic Correlations in UTUC and UBUC

The association of clinicopathological characteristics with *MAP1B* immunoreactivity is shown in Table 2. We found, in UTUC cases, that high *MAP1B* expression was markedly associated with synchronous multiple tumors (*p* = 0.024), advanced pT status (*p* = 0.005) (Figure 2A–C), positive lymph node metastasis (*p* = 0.002), the presence of vascular invasion (*p* < 0.001), and an increased mitotic rate (*p* < 0.001) (Table 2 and Figure 2D). Similarly, in cases with UBUC, we found evidence of associations between increased *MAP1B* expression and advanced pathological tumor stage (*p* < 0.001), positive lymph node metastasis (*p* = 0.012), a high histological tumor grade (*p* = 0.016), the presence of vascular invasion (*p* = 0.045), and an increased mitotic rate (*p* = 0.006) (Table 2 and Figure 2E). Of note, none of the 30 cases displaying high *MAP1B* expression enrolled for mutational analysis were positive for *MAP1B* mutation, suggesting a mutation-independent expression of *MAP1B*.

### 2.3. Survival Analysis in UTUC and UBUC

During follow-up, we found in our UTUC cohort that 61 (17.9%) patients died because of their cancer and 70 (20.6%) patients experienced disease progression. During univariate analysis, we observed that multifocal tumors, advanced pathological tumor stage, positive lymph node metastasis, high histological tumor grade, the presence of vascular invasion, perineural invasion, and high MAP1B expression (Figure 3A,B) were associated with worse disease-specific survival (DSS) and metastasis-free survival (MFS) (all *p* < 0.05). In multivariate analysis, multifocal tumors, advanced pathological tumor stage, positive lymph node metastasis, high histological tumor grade, perineural invasion, and MAP1B expression were independently predictive for both DSS and MFS (all *p* < 0.05) (Table 3).

In our follow-up of UBUC patients, we found that 52 (17.6%) patients died due to the cancer and 76 (25.8%) patients experienced disease progression. During univariate analysis, we determined that advanced pT status, positive lymph node metastasis, high histological tumor grade, the presence of vascular invasion, perineural invasion, an increased mitotic rate, and increment of MAP1B expression (Figure 3C,D) were associated with worse DSS and MFS (all *p* < 0.05). Using multivariate analysis, we confirmed that advanced pathological tumor stage, an increased mitotic rate, and MAP1B expression remained significant in predicting reduced DSS and MFS (all *p* < 0.05) (Table 4).

### 2.4. MAP1B Promotes the Cell Proliferation, Migration, and Invasion of UC Cell Lines

To investigate the biological effects of MAP1B, we first characterized endogenous MAP1B expression in eight UC cell lines and noticed RTCC1 and J82 cells had the most abundant MAP1B transcripts and protein expression (Figure 4A). We next successfully knocked down MAP1B in both the RTCC1 (Figure 4B, left) and J82 (Figure 4B, right) cell lines using short hairpin RNA (shRNA). We found significantly attenuated proliferation (viability) in stable MAP1B-silenced RTCC1 (Figure 4C1) and J82 (Figure 4C2) cells. Due to the positive relationship between MAP1B expression and the development of metastasis, we evaluated the effect of MAP1B in UC cell migration and invasion. MAP1B knockdown significantly decreased the migratory and invasive abilities of RTCC1 (Figure 4C3,C5) and J82 (Figure 4C4,C6) cells.

### 2.5. MAP1B Expression Correlates with Chemoresistance In Vitro and In Vivo

Flow cytometric analysis of stable *MAP1B* knockdown RTCC1 and J82 cell lines showed stable *MAP1B* knockdown significantly increased the sub-G1 population, indicating induced cell apoptosis (Figure 5 and Figure 6). Further analysis of vinblastine-treated RTCC1 and J82 cell lines also disclosed induced cell apoptosis (Figure 7 and Figure 8). In other words, MAP1B expression might lead to a resistance to anti-mitotic chemotherapeutics. In the independent UBUC patient cohort receiving adjuvant chemotherapy, Kaplan–Meier survival analysis showed high MAP1B expression correlated with inferior DFS (Figure 9), further supporting the role of MAP1B in chemoresistance.

## 3. Discussion

It is estimated that one-third of patients with UBUC have advanced disease at presentation [16]. A similar poor prognosis was found among patients with advanced UTUC in that the DSS has not changed significantly during the last two decades [17]. Regardless of the high initial response, the therapeutic effects of current treatment were insufficient and resulted in recurrence and death. Currently, there are no effective salvage regimens for treating metastatic UC. Metastasis requires the inherent dynamic instability of microtubules for cell motility, and many changes in the microtubule network have been identified in various cancers [14]. There is accumulating evidence that MAPs are associated with changes in microtubule dynamics, that they can determine the effects of microtubule-targeting agents, and that they play a role in cancer resistance [14]. However, reliable tumor markers that predict the sensitivity to chemotherapy and resistance to tumor metastasis remain elusive.

MAPs contain products of oncogenes, tumor suppressors, and apoptosis regulators thought to be involved in microtubule assembly. On the other hand, vinblastine, listed in the World Health Organization’s List of Essential Medicines, binds tubulin and inhibits the assembly of microtubules [18]. It causes M-phase–specific cell-cycle arrest by breaking microtubule assembly and proper formation of the mitotic spindle and the kinetochore, which were essential for the separation of chromosomes during the anaphase of mitosis. Due to the possibility of sharing a common function, the rational microtubule-targeting cancer therapeutic approaches should preferably include proteomic profiling of tumor MAPs before the administration of antimicrotubule agents preferentially in combination with agents that modulate the expression of relevant MAPs [14].

Histologically, MAPs were originally related to the development of the nervous system, based on their very early detection in neurons. However, the aberrant expression of primarily neuronal MAPs has since been detected in non-neural cancer tissues [14]. We also assessed MAP1B expression across various cancer types using Oncomine™ Platform (Thermo Fisher, Ann Arbor, MI). Data revealed a diverse expression of MAP1B in various cancers. Of these, CNS tumor has highest MAP1B expression; bladder tumor has moderate expression. In our present results and using a published transcriptome dataset (GSE31684), we first found that *MAP1B* was significantly upregulated in UC and associated with more advanced pT status and metastatic disease in UBUC. Next, we found using immunohistochemistry that *MAP1B* overexpression markedly correlated with disease status in affected patients. In patients with UTUC, *MAP1B* overexpression was positively associated with synchronous multiple tumors, advanced pathological tumor stage, positive lymph node metastasis, the presence of vascular invasion, and an increased mitotic rate. However, in patients with UBUC, *MAP1B* overexpression was associated with advanced pathological tumor stage, positive lymph node metastasis, high histological tumor grade, the presence of vascular invasion, and an increased mitotic rate. Furthermore, using survival analysis, we demonstrated an association between *MAP1B* and aggressive clinical progression, whereby *MAP1B* overexpression independently predicted poor DSS and MFS rates for all patients with UC. These findings indicate that standard clinical practices may benefit from evaluating the *MAP1B* status to improve the risk stratification of patients with UC.

Different *MAP1B* interactors can be grouped into seven different categories, including signaling, cytoskeleton, transmembrane proteins, RNA-binding proteins, apoptosis, neurodegeneration-linked proteins, and neurotransmitter receptors [19]. *MAP1B* is translated as a precursor polypeptide that undergoes proteolytic processing to cleave into an N-terminal heavy chain (*MAP1B* HC) and a C-terminal light chain (*MAP1B* LC1). *MAP1B* LC1 overexpression, which can generate protein aggregates, has been observed in endoplasmic reticulum-related stress-induced cell apoptosis. This effect is blocked by DJ-1, a Parkinson’s disease–related protein that has been proposed to act like a molecular chaperone, and inhibits α-synuclein aggregation [20]. However, in contrast to the proapoptotic effects caused by LC1 overexpression, *MAP1B* overexpression is not related to cell death related to *p53*, a tumor-suppressor gene; in fact, *MAP1B* overexpression reduces *p53* transcriptional activity and inhibits doxorubicin-induced apoptosis [21]. In addition, we found that the percentages of cells in the early and late stages of apoptosis were significantly increased between shLacZ controls and shMAP1B-treated cells. Further in vivo studies are warranted to confirm our findings and to determine whether such results may lead to new therapeutic targets for UC.

Recent studies have found that changes in the expression of MAPs are associated with chemotherapy resistance and cancer progression [14,22]. For example, stathmin plays a role in regulating neuroblastoma cell migration and invasion [22]. Silencing stathmin expression using RNAi gene silencing significantly reduced lung metastasis in neuroblastoma in vivo. Similarly, we demonstrated using UC cell lines with high endogenous *MAP1B* expression that silencing by *MAP1B* shRNA significantly reduced cell proliferation, migration, and invasion ability. Based on these findings, we posit that *MAP1B* may be a clinically valuable diagnostic marker for early cancer detection and a promising prognostic marker.

Further, *MAP1B* interacts with several other proteins associated with cancer. For example, Ras-association domain family 1 isoform A (RASSF1A), a tumor suppressor whose inactivation is implicated in the development of many human cancers, interacts with *MAP1B* to influence microtubule dynamics in the cell cycle and is involved in the inhibition of cancer cell growth [23]. Through distinct bifunctional structural domains, C19ORF5, a sequence homolog of *MAP1B*, mediates the communication between the microtubular cytoskeleton and mitochondria in the control of cell death and defective genome destruction. In addition, it has been proposed that the accumulation of C19ORF5 results in microtubule hyperstability, which may be involved in the tumor suppression activity of RASSF1A [24]. In the mammary cancer susceptibility 1 (*Mcs1*) region in chromosome 2 (a region that expresses centromeric proteins), Laes et al. analyzed candidate genes in the region and found that *MAP1B* was expressed in the mammary glands of rats [25]. Interactions with other proteins not related to its role in stabilizing microtubules suggest that *MAP1B* may be part of a “signaling protein” that regulates molecular pathways [19]. We propose that *MAP1B* has multiple functions, and whether the main function of *MAP1B* is microtubule stabilization or whether it has many cellular functions warrants further investigation.

A recent study that focused on kidney glomerular development and function found that *MAP1B* was specifically expressed in podocytes in human and murine adult kidney tissues [26]. In a mouse model, *MAP1B* was not essential for glomerular filtration function but may play a role in the development and differentiation of the kidney tubular system. The authors hypothesize that *MAP1B* may be related to either stress maintenance or the aging process in the kidney. It is clear that the overall effects of *MAP1B* on UC are complex, with reports of associations between *MAP1B* and survival and metastasis. Research aimed at decoding the functional consequences of *MAP1B* and signaling cross-talk with other proteins in different cancers is needed in the future. However, due to a slight predominance toward females, it is unclear if the results can easily be transferred to the rest of the world.

## 4. Materials and Methods

### 4.1. Data Mining of GSE31684 to Identify Altered Gene Expression in UC

The transcriptome dataset GSE31684 (http://www.ncbi.nlm.nih.gov/geo/query/acc.cgi?acc=GSE31684), which includes 93 patients with UBUC who underwent radical cystectomy, was obtained from the Gene Expression Omnibus repository at the National Center for Biotechnology Information. Raw data were imported by Nexus Expression 3 (BioDiscovery, EI Segundo, CA, USA) to quantify the gene expression level. No pre-selection or filtering was conducted during the analysis of the data for all probes. Comparative analyses were performed to determine the significant differences in the expressed genes by comparing the primary tumor (pT) status (high-stage to low-stage) and the presence or absence of metastatic events.

### 4.2. Patients and Tumor Specimens

Between 1996 and 2004, 340 patients with UTUC and 295 with UBUC who underwent surgery with curative intent at the Chi Mei Medical Center were enrolled. This study was reviewed and approved by the institutional review board (105-01-005). Informed patient consent was obtained from all participants. Demographic characteristics and clinical information including pathological features, oncological follow-up, and cause of mortality were retrospectively collected. Patients who underwent neoadjuvant chemotherapy or radiotherapy; who had concurrent muscle-invasive bladder tumor, acute blood disorders, or bone marrow diseases; and those with incomplete clinical information were excluded from our study. The tumor stage was defined in accordance with the 2002 American Joint Committee Cancer (AJCC)’s Tumor, Node, Metastasis system. Two pathologists reviewed tumor tissues and reclassified then as low- or high-grade using the seventh edition of the AJCC staging system. As a rule, all patients were treated initially by surgery with curative intent. All UBUC patients with pT3 or pT4 diseases or with nodal involvement received cisplatin-based adjuvant chemotherapy. However, of the 106 UTUC patients with pT3 or pT4 and nodal positive diseases, only 29 received cisplatin-based adjuvant chemotherapy. One expert pathologist (CFL) re-evaluated the hematoxylin and eosin–stained sections of all cases. To determine the *MAP1B* transcript level, a pilot batch of 30 UTUC and 30 UBUC snap-frozen tissues with a high tumor percentage (> 70%) was retrieved. Each group included 10 tumor tissues of the pTa stage, 10 of the pT1 stage, and 10 that were muscle-invasive (pT2–pT4).

### 4.3. Immunohistochemical Staining

Immunohistochemistry was conducted to detect *MAP1B* protein expression in 340 UTUC and 295 UBUC cases. One representative slide of a tumor with most invasive area was evaluated by two pathologists manually. Tumor tissue slide preparation was performed as described in our previous study [27]. Slides were incubated with the primary antibody against *MAP1B* (1:100, clone AA6; Millipore, Beverly, MA, USA). We quantified *MAP1B* protein expression levels by combining the intensity and percentage of immunostaining in the cytoplasm of UC cells to generate an H score using the following equation: H score = ΣPi (i + 1), where Pi is the percentage of stained tumor cells (0–100%) and i represents the intensity of immunoreactivity (0–3+). The resulting scores ranged from 100 to 400 points, where a score of 100 points indicated that 100% of cancer cells were nonreactive and a score of 400 points meant that 100% of the cancer cells examined were strongly immunoreactive (3+).

### 4.4. Real-Time Reverse-Transcription Polymerase Chain Reaction (RT-PCR) to Assess the Transcription Levels of MAP1B in Cell Lines and UC Samples

We calculated the fold change in *MAP1B* gene expression of UC tumors relative to that of normal tissues as previously described [27]. We extracted total RNA from cell lines and a pilot batch of cases consisting of 30 UTUCs and 30 UBUCs to quantify the transcription level of *MAP1B* using real-time RT-PCR. Predesigned TaqMan assay reagents (Applied Biosystems, Waltham, MA, USA) were used to assess the mRNA abundance of *MAP1B* (Hs00195485_m1) using the ABI StepOnePlus™ system (Applied Biosystems, Waltham, MA, USA), for which *POLR2A* (Hs01108291_m1) was used as the internal control for normalization.

### 4.5. Cell Culture

The cell lines RT4, TCCSUP, J82, and HUC were purchased from the American Type Culture Collection (Manassas, VA, USA). The cell lines BFTC 909, and BFTC 905 were obtained from the Food Industry Research and Development Institute (Hsinchu, Taiwan). RTCC1 cells were kindly provided by Professor Lien-Chai Chiang at Kaohsiung Medical University [28]. Short-tandem repeat profiling cell authentication had been performed in all cell lines (Mission Biotech, Taipei, Taiwan).

### 4.6. RNA Interference

The lentiviral vectors pLKO.1-*shLacZ* (TRCN0000072223: 5′-TGTTCGCATTATCCGAACCAT-3′) and pLKO.1-*shMAP1B* (#1, TRCN0000116621: 5′-GCCTGGAATAAACAGCATGTT-3′; #2, TRCN0000290688: 5′- CCCTGACTTAGGAGTTGTATT-3′) were obtained from the Taiwan National RNAi Core Facility (Taipei, Taiwan) and used to establish stable *MAP1B*-silenced clones of RTCC1 and J82 cell lines using shRNAs against *MAP1B* (*shMAP1B*).Viruses were produced by transfecting HEK293 cells with the above three vectors using Lipofectamine 2000 (Thermo Fisher Scientific, Waltham, MA, USA) [29]. For viral infection, 3 × 10^6^ RTCC1 and J82 cells were incubated with 8 mL of lentivirus in the presence of polybrene, followed by puromycin selection of the stable clones of lentivirus-transduced cells.

### 4.7. Western Blotting

Our previously published western blotting assay procedure was used to evaluate endogenous *MAP1B* expression and the *MAP1B*-knockdown efficiency in RTCC1 and J82 cell lines using primary antibodies against *MAP1B* (1:500, clone AA6; Millipore, Beverly, MA, USA) and glyceraldehyde 3-phosphate dehydrogenase (GADPH) (6C5, 1:10,000; Millipore, Beverly, MA, USA). Cell lysates with 25 μg of protein were separated using a 4% to 12% gradient NuPAGE gel (Invitrogen, Carlsbad, CA, USA), then transferred onto polyvinylidene difluoride membranes (Amersham Biosciences, Buckinghamshire, UK) for the immobilization of proteins. Membranes were incubated with tris-buffered saline containing Tween 20 (TBST) buffer and 5% skimmed milk at room temperature for one hour for blocking, followed by exposure to primary antibodies at 4 °C overnight against *MAP1B* (1:500, clone AA6; Millipore, Beverly, MA, USA) using GADPH as a loading control (6C5, 1:10,000; Millipore, Beverly, MA, USA). Membranes were incubated with the secondary antibody at room temperature for 1.5 h, and proteins were detected using a chemiluminescence system (Amersham Biosciences, Buckinghamshire, UK).

### 4.8. Bromodeoxyuridine (BrdU) Assay to Assess DNA Synthesis

DNA synthesis was measured using an enzyme-linked immunosorbent assay (ELISA)-based and colorimetric bromodeoxyuridine (BrdU) assay (Roche Holding AG, Basel, Switzerland). *MAP1B*-knockdown or *shLacA* control RTCC1 and J82 cell lines were plated into a 96-well plate at a density of 3000 cells per well. At 24, 48, and 72 h, we measured the amount of DNA synthesis. The labeling medium was removed after three hours of incubation with BrdU at 37 °C under 5% CO_2_, followed by fixation and a final incubation with an anti-BrdU-POD solution. An ELISA reader (Promega Corp., Madison, WI, USA) was used to measure the absorbance at 450 nm, and the reference was set at an absorbance of 690 nm.

### 4.9. Pharmacological Assays

The colorimetric 2,3-bis-(2-methoxy-4-nitro-5-sulfophenyl)-2H-tetrazolium-5-carboxanilide (XTT) assay (Sigma-Aldrich, St. Louis, MO, USA) was used to assess cell viability as previously described [30]. Vinblastine sulfate (Hospira UK Ltd., Maidenhead, UK) was obtained and suspended in normal saline. RTCC1 and J82 cells were seeded in 96-well plates at a density of 5 × 10^3^ cells per well the day before treatment at the indicated time points with vehicle control (0.9% saline) or increasing concentrations of vinblastine sulfate. The length of treatment interval was 72 h. After incubation with XTT reaction mixture for three hours at 37 °C under 5% CO_2,_ the absorbance of the samples was determined using an ELISA reader (Promega Corp., Madison, WI, USA) at 450 nm, with the absorbance set at 630 nm as reference.

### 4.10. Migration and Invasion Assays

Cell migration assay was performed using Falcon HTS FluoroBlok 24-well inserts (BD Biosciences, Franklin Lakes, NJ, USA) and the cell invasion assay was performed using the 24-well Collagen-based Cell Invasion Assay (Millipore, Beverly, MA, USA). Briefly, we added serum-free medium to rehydrate each insert, then replaced it with a serum-free suspension with equal numbers of cells in the upper chamber, followed by a 12- to 24-h incubation period to allow cells to migrate toward (i.e., invade) the lower chamber, which contained medium with 10% fetal bovine serum. After removal of the noninvading cells in the upper chamber, cells that invaded through the inserts were stained, lysed in extraction buffer, and transferred to 96-well plates for colorimetric readings at 560 nm.

### 4.11. Flow Cytometry Analysis of Cell-Cycle Kinetics

Stable pools of *MAP1B* knockdown versus the corresponding *shLacZ* control of the RTCC1 and J82 cell lines were pelleted and fixed overnight in 75% cold ethanol at −20 °C. The cells were washed twice using cold phosphate-buffered saline with 10 mg/mL of DNase-free RNase. Next, the cells were labeled with 0.05 mg/mL of propidium iodide and analyzed using a NovoCyte flow cytometer (ACEA Biosciences, San Diego, CA, USA) to determine the different proportions of cells at each phase of the cell cycle. Our lower limit of the number of sorted cells after gating out fixation artifacts and cell debris was 10^4^ cells for all experiments.

### 4.12. Flow Cytometry Analysis of Apoptosis

Cell apoptosis was evaluated by plating RTCC1 and J82 cells (10^5^ cells each) with sh*LacZ* or sh*MAP1B* for 24 h, followed by 15 min of incubation using an Annexin V-FITC kit (BD Biosciences, Franklin Lakes, NJ, USA) that contained propidium iodide. The percentages of cells at late apoptosis were calculated from three independent experiments.

### 4.13. Mutation Analysis

To explore potential *MAP1B* mutation in UC, we randomly selected 15 UTUC and 15 UBUC cases (Appendix A) with high protein expressions of *MAP1B* for mutation analysis. Mutation analyses were performed by using an ABI3100 sequencer targeting eight pathogenic point mutations occurring in other cancer types according to the database of COSMIC repository (https://cancer.sanger.ac.uk/cosmic/gene/analysis?ln=HSD11B1#variants). Validated *MAP1B* mutations and primers sets are shown in Appendix A. The PCR amplification started with an initial denaturation step at 95 °C for 15 min, followed by 35 cycles of 95 °C for 30 s, 58 °C for 30 s, and 72 °C for 30 s, and a final extension step at 72 °C for 10 min. Then, these amplicons generated in individual PCR reactions were analyzed by direct sequencing.

### 4.14. Postoperative Adjuvant Chemotherapy in UBUC

To evaluate the role of *MAP1B* expression in the response to adjuvant chemotherapy in UBUC patients, an independent cohort containing 70 patients with pT3 or pT4 disease or with nodal involvement received cisplatin-based adjuvant chemotherapy combined with vinblastine and were enrolled for further survival analysis (Appendix A).

### 4.15. Statistical Analyses

The Statistical Package for the Social Sciences version 12.0 software program (IBM Corp., Armonk, NY, USA) was used for all statistical analyses. Differences between categorical parameters were assessed using the chi-squared or Fisher’s exact test. The median H scores of *MAP1B* immunoreactivity were used as cutoff values to separate UTUC and UBUC into two subgroups of high and low *MAP1B* expression. Pearson’s chi-squared test was used to compare the association between *MAP1B* expression and clinicopathological parameters. The Kaplan–Meier method was applied to estimate the effect of *MAP1B* expression on DSS and MFS. The survival curves were compared using the log-rank test. We used a Cox proportional-hazards model to identify independent predictors for DSS and MFS. In all figure legend, continuous parameters (such as MAP1B transcript expression in Figure 1, mitotic activity in Figure 2, *MAP1B* mRNA expression, relative proliferation, migration and invasion in Figure 4, apoptosis rate in Figure 6) were assessed using a t-test or Mann–Whitney–Wilcoxon test. Survival analysis (DSS and MFS) were performed using Kaplan-Meier plots and compared by the log-rank test. Statistical significance was set at *p* < 0.05.

## 5. Conclusions

In summary, the present study demonstrated that MAP1B overexpression was not only an indicator of unfavorable clinicopathological parameters, but also an independent prognostic factor able to predict poor DSS and MFS rates in patients with UTUC or UBUC. Additional studies must be conducted to elucidate the details of the biological significance of MAP1B and its encoded protein in UC oncogenesis for exploring possible MAP1B-targeted therapy for both kinds of UC.

## Figures and Tables

**Figure 1 cancers-12-00630-f001:**
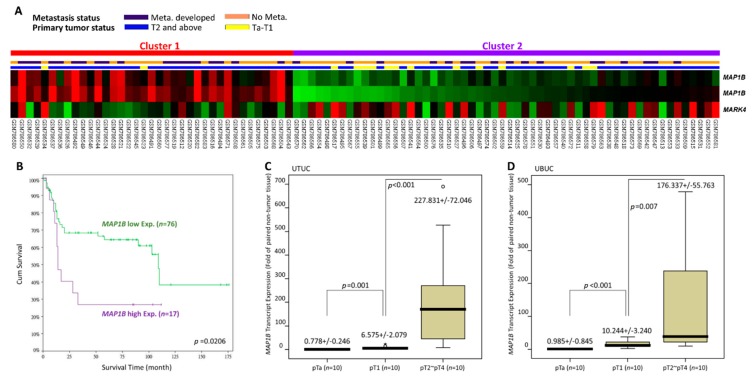
Analysis of gene expression in urinary bladder urothelial carcinoma (UBUC) using a published transcriptome dataset (GSE31684). (**A**) Cluster analysis of genes focusing on the GO microtubule bundle formation class (GO:0001578) revealed that *MAP1B* was one of the most significantly upregulated genes associated with more advanced pT status and metastatic disease. Tissue specimens from cancers with a distinct pT status are illustrated at the top of the heat map, and the expression levels of upregulated and downregulated genes are represented as a continuum of brightness of red or green, respectively. Specimens with no change in messenger RNA (mRNA) expression are shown in black. (**B**) Kaplan–Meier plots showing the prognostic significance of *MAP1B* expression for the survival of UBUC. Using a QuantiGene assay, *MAP1B* mRNA expression was significantly increased in both (**C**) upper tract urothelial carcinoma (UTUC) and (**D**) UBUC at advanced primary pT stages.

**Figure 2 cancers-12-00630-f002:**
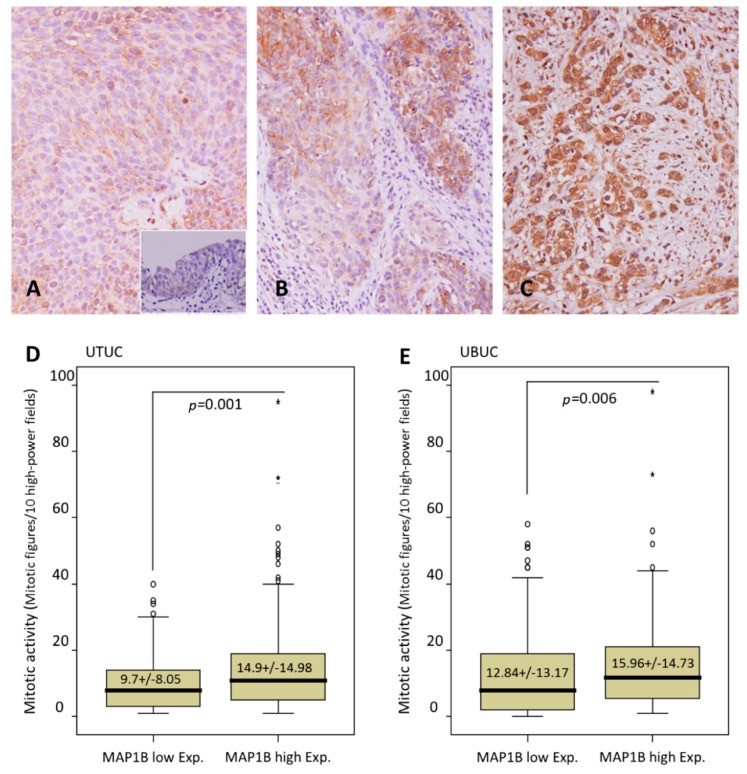
Representative sections of *MAP1B* immunostaining. Note the stepwise increments in *MAP1B* immunoreactivity from the nontumoral urothelial epithelium (inlet) and (**A**) noninvasive papillary UCs to (**B**) non–muscle-invasive (pT1), and (**C**) muscle-invasive (pT2–pT4) UCs. A comparison of mitotic activity showed significantly higher mitotic rates in (**D**) UTUC and (**E**) UBUC cells with increased *MAP1B* expression than in cells with low expression.

**Figure 3 cancers-12-00630-f003:**
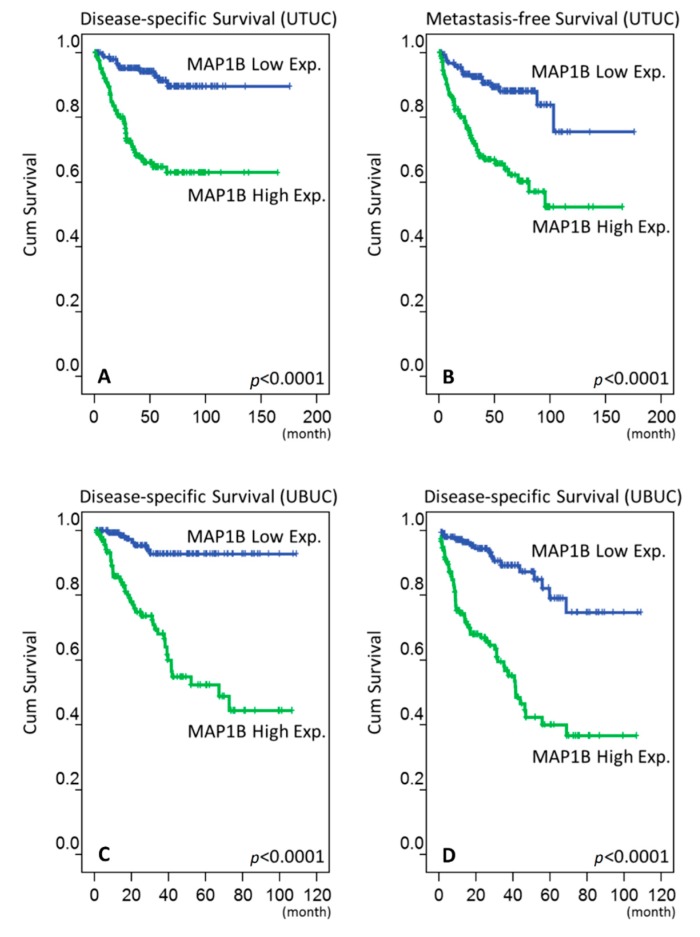
Kaplan–Meier survival analysis showing the prognostic significance of MAP1B expression for the DSS and MFS outcomes of UTUC (**A** and **B**) and UBUC (**C** and **D**).

**Figure 4 cancers-12-00630-f004:**
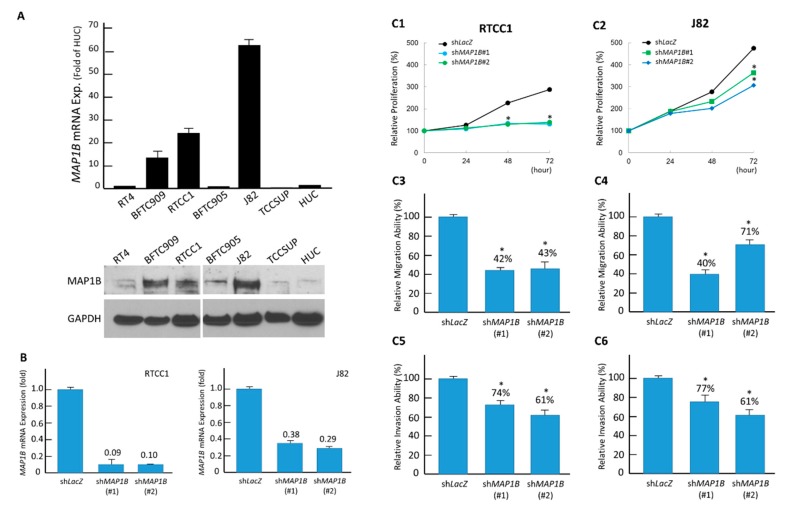
*MAP1B* expression promotes the growth of UC cells in vitro. (**A**) As compared with RT4 cells, endogenous *MAP1B* mRNA (upper) and protein (lower) expressions were increased in cells from the J82 and RTCC1 cell lines. (**B**) The two cell lines with high endogenous *MAP1B* expression were stably silenced against *MAP1B* expression by a lentiviral vector bearing one of the two clones of *MAP1B* shRNA with different sequences for both RTCC1 (left panel) and J82 (right panel) cells. Using an ELISA-based colorimetric assay to assess the rate of BrdU uptake, cell proliferation was significantly reduced in stable *MAP1B*-knockdown (**C1**) RTCC1 and (**C2**) J82 cell lines compared with that in the corresponding shLacZ controls. Similar trends were found for cell migration and invasion among cells from the (**C3** and **C5**) RTCC1 and (**C4** and **C6**) J82 cell lines. (* *p*<0.05). More details of western blot, please view at the Appendix A.

**Figure 5 cancers-12-00630-f005:**
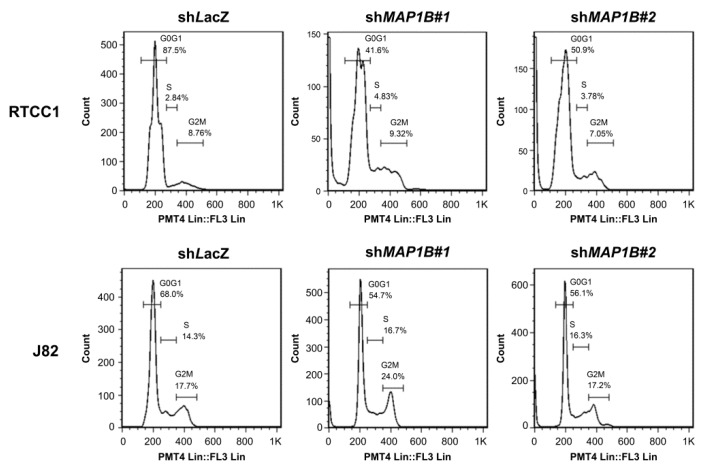
Stable *MAP1B* knockdown increases the sub-G1 population with significantly altered cell-cycle progression. Cell-cycle analysis as conducted by flow cytometry identified a remarkable increment of sub-G1 population indicating cell death in *MAP1B*-knockdown RTCC1 (upper panel) and J82 (lower panel) cells.

**Figure 6 cancers-12-00630-f006:**
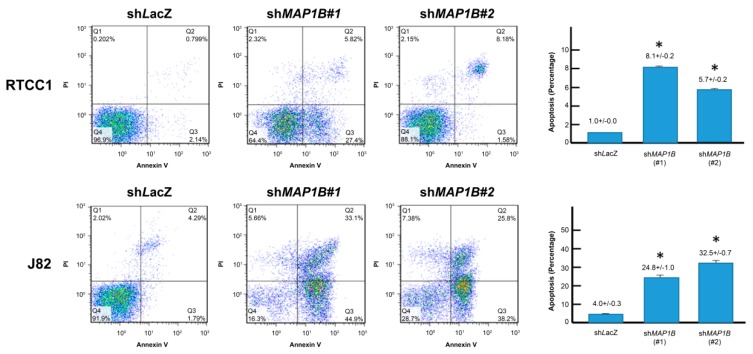
*MAP1B* knockdown induces apoptosis. Flow cytometric analysis of annexin V/propidium iodide-stained RTCC1 (upper panel) and J82 (lower panel) cell lines disclosed *MAP1B* knockdown significantly increased percentage of apoptosis. (* *p* < 0.05).

**Figure 7 cancers-12-00630-f007:**
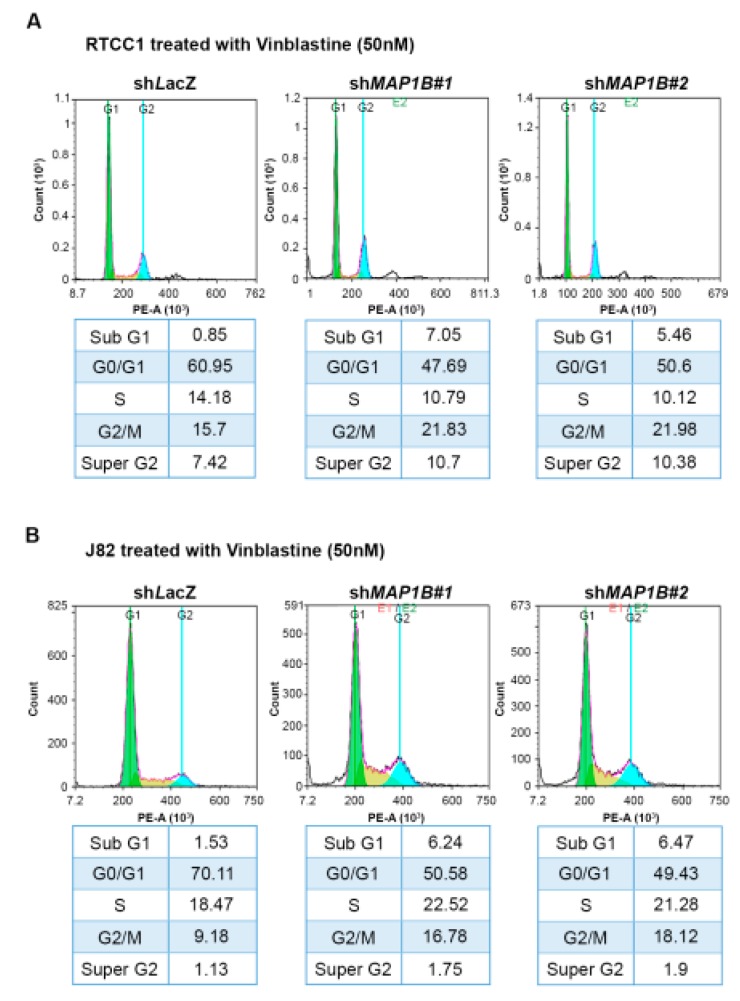
Stable *MAP1B* knockdown increased vinblastine-induced apoptosis. Flow cytometric analysis of vinblastine-treated RTCC1 (upper panel) and J82 (lower panel) cell lines disclosed that *MAP1B* knockdown significantly increased the sub-G1 population, indicating induced cell apoptosis.

**Figure 8 cancers-12-00630-f008:**
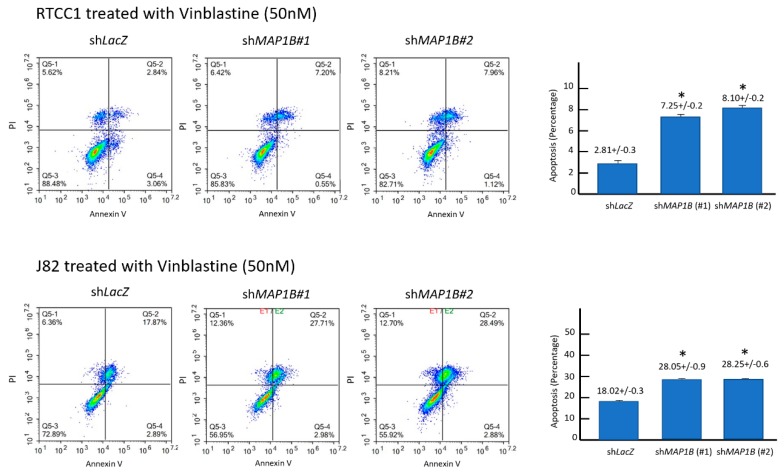
Stable *MAP1B* knockdown increase vinblastine-induced apoptosis. Flow cytometric analysis of annexin V/propidium iodide-stained RTCC1 (upper panel) and J82 (lower panel) cell lines demonstrated *MAP1B* knockdown significantly increased percentage of vinblastine-induced apoptosis. (* *p* < 0.05).

**Figure 9 cancers-12-00630-f009:**
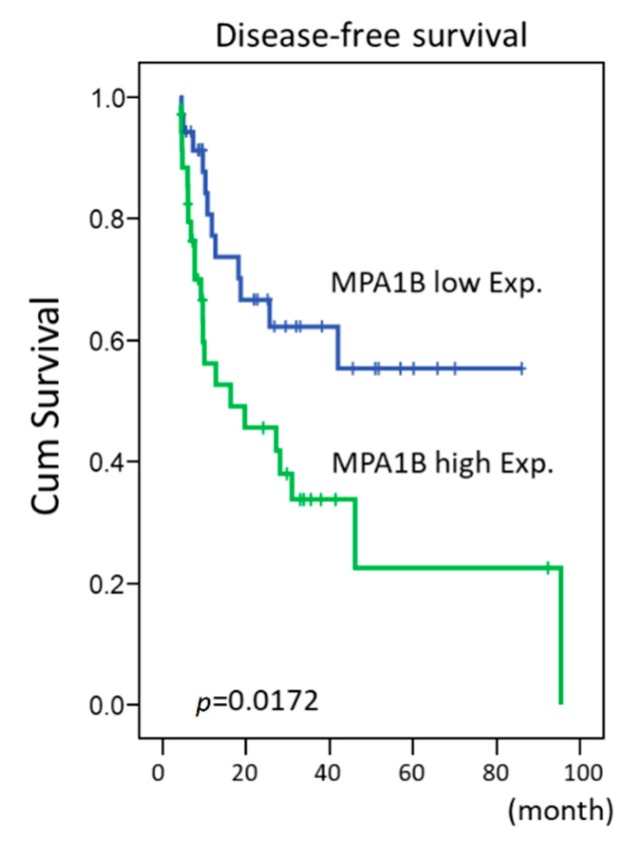
Kaplan–Meier survival analysis of *MAP1B* expression for the DFS of the UBUC patient cohort receiving adjuvant chemotherapy. Kaplan–Meier survival analysis showing the prognostic significance of *MAP1B* expression for the DFS of the UBUC patient cohort receiving adjuvant chemotherapy.

**Table 1 cancers-12-00630-t001:** Summary of differentially expressed genes associated with microtubule bundle formation (GO: 0001578) and showing positive associations to cancer invasiveness and metastasis in the transcriptome of UBUC (GSE31684).

Probe Title 1	Comparing T2-4 to Ta-T1		Comparing Meta. to Non-Meta. ^#^		Gene Symbol	Gene Title	Biological Process	Molecular Function
	Log ratio	*p*-Value	Log ratio	*p*-Value				
214577_at	0.3773	0.0029	0.3943	<0.0001	*MAP1B*	Microtubule-associated protein 1B	Dendrite development, microtubule bundle formation	Protein binding, structural molecule activity
221560_at	−0.3436	0.0058	−0.0115	0.9048	*MARK4*	MAP/microtubule affinity-regulating kinase 4	G1/S transition of mitotic cell cycle, G2/M transition of mitotic cell cycle, Wnt receptor signaling pathway, microtubule bundle formation, microtubule cytoskeleton organization and biogenesis, nervous system development, positive regulation of cell proliferation, positive regulation of programmed cell death, protein amino acid phosphorylation	ATP binding, gamma-tubulin binding, kinase activity, microtubule-binding, nucleotide-binding, protein-binding, protein kinase activity, protein serine/threonine kinase activity, protein-tyrosine kinase activity, tau-protein kinase activity, transferase activity, ubiquitin-binding
226084_at	1.2832	<0.0001	0.9436	<0.0001	*MAP1B*	Microtubule-associated protein 1B	Dendrite development, microtubule bundle formation	Protein-binding, structural molecule activity

#, Meta., distal metastasis developed during follow-up; Non-Meta.: no metastatic event developed.

**Table 2 cancers-12-00630-t002:** Correlations between *MAP1B* expression and other important clinicopathological parameters in UCs.

Parameter	Category	Upper Urinary Tract Urothelial Carcinoma				Urinary Bladder Urothelial Carcinoma			
		Case no.	*MAP1B* Expression		*p*-value	Case no.	*MAP1B* Expression		*p*-value
			Low	High			Low	High	
Gender ^&^	Male	158	79	79	1.000	216	103	113	0.223
	Female	182	91	91		79	44	35	
Age (years) ^#^		340	65.2+/−9.87	65.9+/−9.92	0.409	295	65.76+/−12.02	66.33+/−12.44	0.759
Tumor location	Renal pelvis	141	64	77	0.023 *	-	-	-	-
	Ureter	150	87	63		-	-	-	-
	Renal pelvis & ureter	49	19	30		-	-	-	-
Multifocality^&^	Single	278	144	134	0.160	-	-	-	-
	Multifocal	62	26	36		-	-	-	-
Primary tumor (T) ^&^	Ta	89	54	35	0.005 *	84	56	28	<0.001 *
	T1	92	5`	41		88	45	43	
	T2	159	65	94		123	46	77	
Nodal metastasis ^&^	Negative (N0)	312	164	148	0.002 *	266	139	127	0.012 *
	Positive (N1–N2)	28	6	22		29	8	21	
Histological grade ^&^	Low grade	56	34	22	0.079	56	36	20	0.016 *
	High grade	284	136	148		239	111	128	
Vascular invasion ^&^	Absent	234	132	102	<0.001 *	246	129	117	0.045 *
	Present	106	38	68		49	18	31	
Perineural invasion ^&^	Absent	321	162	159	0.479	275	140	135	0.169
	Present	19	8	11		20	7	13	

&, Chi-squared test; #, Mann–Whitney U test; * Statistically significant.

**Table 3 cancers-12-00630-t003:** Univariate log-rank and multivariate analyses for DSS and MFS in UTUC.

Parameter	Category	Case No.	Disease-Specific Survival					Metastasis-Free Survival				
			Univariate Analysis		Multivariate Analysis			Univariate Analysis		Multivariate Analysis		
			No. of Event	*p*-value	R.R.	95% C.I.	*p*-value	No. of Event	*p*-value	R.R.	95% C.I.	*p*-value
Gender	Male	158	28	0.8730	-	-	-	32	0.8307	-	-	-
	Female	182	33		-	-	-	38		-	-	-
Age (years)	<65	138	26	0.9728	-	-	-	30	0.8667	-	-	-
	≥65	202	35		-	-	-	40		-	-	-
Tumor side	Right	177	34	0.7188	-	-	-	38	0.3903	-	-	-
	Left	154	26		-	-	-	32		-	-	-
	Bilateral	9	1		-	-	-	0		-	-	-
Tumor location	Renal pelvis	141	24	0.0100 *	1	-	0.562	31	0.0752	-	-	-
	Ureter	150	22		1.167	0.618–2.203		25		-	-	-
	Renal pelvis & ureter	49	15		1.261	0.345–4.615		14		-	-	
Multifocality	Single	273	48	0.0031 *	1	*-*	0.050 *	52	0.0144 *	1	-	0.001 *
	Multifocal	62	18		2.238	0.998–5.017		18		2.648	1.496–4.687	
Primary tumor (T)	Ta	89	2	<0.0001 *	1	-	0.008 *	4	<0.0001 *	1	-	0.036 *
	T1	92	9		2.641	0.561–12.419		15		2.643	0.563–12.410	
	T2–T4	159	50		5.667	1.250–25.699		51		5.538	1.236–24.817	
Nodal metastasis	Negative (N0)	312	42	<0.0001 *	1	-	<0.001 *	55	<0.0001 *	1	-	<0.001 *
	Positive (N1–N2)	28	19		4.188	2.244–7.819		15		4.421	2.415–8.094	
Histological grade	Low	56	4	0.0177 *	1	-	0.008 *	3	0.0022 *	1	-	0.008 *
	High	284	57		4.746	1.514–14.881		67		4.770	1.509–15.077	
Vascular invasion	Absent	234	24	<0.0001 *	1	-	0.139	26	<0.0001 *	1	-	0.147
	Present	106	37		1.571	0.863–2.859		44		1.565	0.855–2.868	
Perineural invasion	Absent	321	50	<0.0001 *	1	-	<0.001 *	61	<0.0001 *	1	-	<0.001 *
	Present	19	11		4.768	2.251–10.102	-	9		4.865	2.294–10.318	
Mitotic rate (per 10 high power fields)	<10	173	27	0.1442	-	-	-	30	0.0739	-	-	-
	≥10	167	34		-	-	-	40		-	-	-
*MAP1B* expression	Low	170	11	<0.0001 *	1	-	0.001 *	17	<0.0001 *	1	-	<0.001 *
	High	170	50		4.115	2.077–8.154		53		3.962	2.022–7.763	

* Statistically significant.

**Table 4 cancers-12-00630-t004:** Univariate log-rank and multivariate analyses for DSS and MFS in UBUC.

Parameter	Category	Case No.	Disease-Specific Survival					Metastasis-Free Survival				
			Univariate Analysis		Multivariate Analysis			Univariate Analysis		Multivariate Analysis		
			No. of Event	*p*-value	R.R.	95% C.I.	*p*-value	No. of Event	*p*-value	R.R.	95% C.I.	*p*-value
Gender	Male	216	41	0.4404	-	-	-	60	0.2786	-	-	-
	Female	79	11		-	-	-	16		-	-	-
Age (years)	<65	121	17	0.1010	-	-	-	31	0.6285	-	-	-
	≥65	174	35		-	-	-	45				
Primary tumor (T)	Ta	84	1	<0.0001 *	1	-	<0.001 *	4	<0.0001 *	1	-	<0.001 *
	T1	88	9		6.493	0.696–60.560		23		5.044	1.469–17.327	
	T2–T4	123	42		27.783	3.011–256.370		49		7.845	2.239–27.484	
Nodal metastasis	Negative (N0)	266	41	0.0001 *	1	-	0.729	61	<0.0001 *	1	-	0.100
	Positive (N1–N2)	29	11		1.132	0.560–2.288		15		1.685	0.905–3.137	
Histological grade	Low grade	56	2	0.0010 *	1	-	0.714	5	0.0005*	1	-	0.572
	High grade	239	50		0.744	0.153–3.610		71		0.729	0.244–2.179	
Vascular invasion	Absent	246	37	0.0017 *	1	-	0.174	54	0.0001 *	1	-	0.798
	Present	49	15		0.624	0.316–1.231		22		1.083	0.590–1.985	
Perineural invasion	Absent	275	44	<0.0001 *	1	-	0.099	66	0.0006 *	1	-	0.339
	Present	20	8		2.990	0.878–4.510		10		1.422	0.690–2.930	
Mitotic rate (per 10 high power fields)	<10	139	12	<0.0001 *	1	-	0.021 *	23	<0.0001 *	1	-	0.045 *
	≥10	156	40		2.184	1.124–4.246		53		1.697	1.012–2.846	
*MAP1B* expression	Low	147	7	<0.0001 *	1	-	<0.001 *	16	<0.0001 *	1	-	<0.001 *
	High	148	45		5.551	2.466–12.498		60		3.770	2.146–6.622	

* Statistically significant.

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
