# Peer review of "Role of Microtubule-Associated Protein 1b in Urothelial Carcinoma: Overexpression Predicts Poor Prognosis"

_cancers, 2020, doi:10.3390/cancers12030630_

Round 1

Reviewer 1 Report

Ciels and colleagues describe the impact of Mircotubulus associated MAP1B in urothelial carcinoma in a Taiwan patient cohort. The study indicates that a MAP1B overexpression is associated with adverse clinical features and independetly predicts unfavourable prognostic effects. The manuscript is written well and sufficient information is provided on the different results.  However, the examinations were performed in Taiwan patients only. Of note, in the introduction the authors describe an increased number of upper urinary tract carcinomas and a relatively higher percentage of females in this patient population. Therefrore, it is unclear if the results can easily be transfered to the rest of the world. This aspect should be included in the discussion section.

Unfortunatelly, only few experiments were carried out to clarify the exact mechanism of action of MAP1B. Further information would be of high impact to better clarify the role of MAP1B in course of disease and devolment of resistance of RCC.

Minor comments: vinblastine cannot be introduced as "one of the most effective and safe medicines needed in a health system". In fact, it is mostly applied in combinational treatment regimens because of an insufficient activity as a single drug. Side effects include myelosuppression, neurotoxicity, obstipation, infections etc.

Reviewer 2 Report

Chien et al. manuscript identified that microtubule-associated protein 1b (MAP1B) served as a predictive factor for bladder cancer patients. The authors found that MAP1B was significantly up-regulated in urothelial carcinoma of the bladder (UBUC) from published transcriptome dataset. They further examined the oncogenic role of MAP1B on cell proliferation, migration and invasion. However, there are several unclarified issues waiting for elucidation.

Major concerns

The authors found that upregulated gene level of MAP1B in UBUC was associated with poor prognosis and more aggressive phenotype from published transcriptome dataset. Do authors analyze gene level of MAP1B from TCGA database, the biggest cohort of localized tumors, or other published transcriptome datasets? To strengthen the role of MAP1B in UBUC, it is strongly encouraged to include several public domain data. Knockdown MAP1B suppressed cell proliferation, migration and invasion of RTCC1 and J82 cell lines. Whether MAP1B directly regulates these aggressive properties or not. Knockdown MAP1B in RTCC1 and J82 increased vinblastine-induced apoptosis; it is unclear how microtubule agent targets cancer cells with low expression level of MAP1B? Targeting microtubule in cancer cells is widely applied. Do authors compare expression level of MAP1B between bladder cancer and other cancer types?

Minor concern:

The third row of abstract has an obvious error.

Reviewer 3 Report

Please see file in attachment

Round 2

Reviewer 1 Report

The article can can be accepted in the current version since most of the points were addressed by the authors. I am looking forward to read more about the mechanism of action in the future.

Author Response

To dear Reviewer

Thank you very much for your prudent review and comments on our manuscript.

Sincerely,

Chien-Feng Li, M.D., Ph.D

Address: Chi-Mei Medical Center, Tainan 701, Taiwan

Telephone number: +886-7- 3208212

Fax number: +886-7- 3211033

E-mail address: angelo.p@yahoo.com.tw

Reviewer 2 Report

The revised version of manuscript seems better than old version, and the authors have provided appropriate interpretations on reviewer's questions as much as possible. I have no questions to ask further.

Author Response

To dear Reviewer

Thank you very much for your prudent review and comments.

Sincerely,

Chien-Feng Li, M.D., Ph.D

Address: Chi-Mei Medical Center, Tainan 701, Taiwan

Telephone number: +886-7- 3208212

Fax number: +886-7- 3211033

E-mail address: angelo.p@yahoo.com.tw

Reviewer 3 Report

Please see my comments in attachment
